# Comparison of Titanium and PEEK Medical Plastic Implant Materials for Their Bacterial Biofilm Formation Properties

**DOI:** 10.3390/polym14183862

**Published:** 2022-09-15

**Authors:** Sonia Sarfraz, Pilvi-Helinä Mäntynen, Marisa Laurila, Sami Rossi, Junnu Leikola, Mika Kaakinen, Juho Suojanen, Justus Reunanen

**Affiliations:** 1Biocenter Oulu & Cancer and Translational Medicine Research Unit, University of Oulu, 90014 Oulu, Finland; 2Päijät-Häme Joint Authority for Health and Wellbeing, Department of Oral and Maxillofacial Surgery, Lahti Central Hospital, 15850 Lahti, Finland; 3Cleft Palate and Craniofacial Centre, Department of Plastic Surgery, Helsinki University Hospital, 00029 Helsinki, Finland; 4Oulu Centre for Cell-Matrix Research, Faculty of Biochemistry and Molecular Medicine, University of Oulu, 90220 Oulu, Finland

**Keywords:** *Staphylococcus aureus*, *Streptococcus mutans*, *Enterococcus faecalis*, *Escherichia coli*, titanium, PEEK, bacterial adhesion, saliva

## Abstract

This study investigated two of the most commonly used CAD–CAM materials for patient-specific reconstruction in craniomaxillofacial surgery. The aim of this study was to access the biofilm formation of *Staphylococcus aureus*, *Streptococcus mutans*, *Enterococcus faecalis*, and *Escherichia coli* on titanium and PEEK medical implant materials. Two titanium specimens (titanium grade 2 tooled with a Planmeca CAD–CAM milling device and titanium grade 5 tooled with a computer-aided design direct metal laser sintering device (CAD-DMLS)) and one PEEK specimen tooled with a Planmeca CAD–CAM milling device were studied. Bacterial adhesion on implants was evaluated in two groups (saliva-treated group and non-saliva-treated group) to imitate intraoral and extraoral surgical routes for implant placement. The PEEK medical implant material showed higher bacterial adhesion by *S. aureus*, *S. mutans*, and *E. coli* than titanium grade 2 and titanium grade 5, whereas *E. faecalis* showed higher adhesion to titanium as compared to PEEK. Saliva contamination of implants also effected bacterial attachment. Salivary coating enhanced biofilm formation by *S. aureus*, *S. mutans*, and *E. faecalis*. In conclusion, our findings imply that regardless of the implant material type or tooling techniques used, salivary coating plays a vital role in bacterial adhesion. In addition, the majority of the bacterial strains showed higher adhesion to PEEK than titanium.

## 1. Introduction

Modern medicine together with biomedical engineering has developed the tools to achieve patient-specific, post-traumatic reconstruction and aesthetic surgeries of craniofacial, dental, and spinal defects, as well as surgical oncology [1]. For internal fixation, the necessary requirements for an implant material are long-term stability, biocompatibility, sufficient strength, and the ability to be sterilized without degradation [2].

Metal implants have been widely used in clinical practice for their excellent friction resistance and mechanical strength. However, metal implants have limitations as their elastic modulus does not match with that of normal human bone tissue, which can cause a stress shielding effect on the peri-implant bone, ultimately leading to loosening of the prosthetic implant. Long-term existence of the metal implant can also trigger allergic reactions in the surrounding tissues. Yet a factor that limits the efficiency of metal implants is the radiopacity of metals, which creates artefacts in imaging and interferes with the interpretation of results of the computerized tomography (CT) and magnetic resonance imaging (MRI) scanning of the patients [3].

PEEK (polyetheretherketone) is a newly developed polymer that belongs to the family of PAEKs (polyaryletherketones), which are semi-crystalline thermoplastics with a chemical structure that provides high temperature stability and mechanical strength. PEEK was developed in 1978 and was initially used in industrial applications; however, later in the 1990s, it emerged as a high-performance thermoplastic implantable contender having the potential to replace metal implant materials [4]. In addition to the stable chemical and physical properties of PEEK, it exhibits excellent biocompatibility properties. It does not cause allergic or toxic effects; most importantly, the elastic modulus of PEEK (8.3 GPa) is much closer to that of normal human bone (17.7 GPa) than metal alloys (116 GPa) [3].

Clinical applications of PEEK started in the mid-1980s, when orthopedic surgeons began exploring its use in hip and trauma surgery. PEEK possesses a number of qualities that make it an excellent implant material for trauma and accident applications, and it is commonly used as an orthopedic medical material for support and bone repair [5]. Having the biomechanical properties of PEEK close to human bone reduces the risk of osteolysis and bone resorption caused by the implant stress shielding effect [6]. Because of its physical structure and high performance, PEEK has been used as a metal-free implant material in dental prostheses, implant abutments, healing abutments, dental implants, and others in dentistry [7].

PEEK biomaterials are now accepted as successful implant materials in load-sharing fusion spine applications [8]. PEEK cages have been effectively used in cervical and lumber spinal fusions. The clinical outcome with PEEK cages revealed no implant failure, a significantly higher rate of fusion, and a lower rate of complications as compared with titanium cages. However, to increase the osseointegration of the PEEK cages, the coating of PEEK with bioactive ceramic materials such as bioglass, HA (hydroxyapatite), and β-tricalcium phosphate has been used [9].

The most significant feature of PEEK is radiolucency, which makes it invisible on X-ray, CT, and MRI scans, allowing better visualization of the bone tissue by radiography. This feature is especially useful when the prosthesis covers a large surface on the bone in anatomical regions [2]. Custom-made implants of PEEK have been used for the treatment of craniofacial defects. Compared to titanium, the elasticity modulus and formability of PEEK make it suitable for use in craniofacial and orbital floor defects. Craniofacial defects tend to carry functional damage, physiological consequences, and aesthetic deformities in patients [10]. Different biomaterials such as titanium, autologous bone grafts, and polymethyl methacrylate (PMMA) are being used for cranio-orbital defects; however, for larger reconstructions and complex defects, PEEK is one of the best options [10,11]. A study conducted on the optimization of fused deposition modeling to print PEEK suggested that the Taguchi method helps to enhance the printing of PEEK to obtain the maximum strength. The Taguchi method has special orthogonal arrays to access the factors influencing the design using the minimum number of tests [12]. Another study explained the mechanisms of 3D printing technology to produce dental prosthesis. These technologies have been used for the manufacturing of complex designs of high-accuracy prostheses with a varied range of implant materials such as metals, ceramics, and polymers [13]. Figure 1 demonstrates common clinical applications for CAD–CAM patient-specific reconstruction in craniofacial surgery. The mechanical and CAM-based properties of PEEK and titanium are briefly compared in Table 1.

There are only few studies that have discussed biofilm formation on PEEK in comparison with titanium implants. The aim of this study was to evaluate and compare the bacterial adhesion of commonly isolated bacterial species in implant-related infections on PEEK plastic implant material and the most commonly used titanium implants, Ti grade 2 and Ti grade 5 (Ti–6Al–4V). Bacterial biofilm formation was studied here with and without saliva treatment to evaluate the effect of saliva on bacterial adhesion. This plays a vital role in craniofacial region where an intraoral approach may be used to avoid visible scarring.

## 2. Materials and Methods

### 2.1. PEEK and Titanium Implant Preparation

PEEK and titanium implants used in this study were from our CAD–CAM technical provider Planmeca Ltd. (Helsinki, Finland). All implants were manufactured as 20 mm discs, sterilized, and destained in a similar way as normally provided for the needs of the patients. The implant types studied were PEEK, titanium grade 2, and titanium grade 5 [14].

### 2.2. Saliva Contamination

Bacterial adhesion on implants was carried out in two groups: saliva-treated group and non-saliva-treated group. In the saliva-treated group, saliva contamination of the implants was performed using sterile saliva. Paraffin wax-stimulated saliva was collected from healthy volunteers (Ethics committee of the Northern Ostrobothnia Hospital District, EETTMK 11/2019). Collected saliva was pooled and filtered using a 0.45 µm filter (#167-0045 Nalgene™ Rapid-Flow™ Sterile Single-Use Vacuum Filter Units, Nalgene^®^ 295-4545, Edo, de México, Mexico). The filtered saliva was stored at −80 °C and diluted 1:1 in 1× phosphate-buffered saline (PBS) before the procedure. The pH of saliva was measured before the addition of buffer (7.6) and after the addition of buffer (7.5), residing within the range of the normal pH of saliva (6.2–7.6). Autoclaved PEEK and titanium specimens were first treated in saliva solution for 30 min at room temperature and then washed three times with 1× PBS.

### 2.3. Bacterial Suspension Preparation and Biofilm Formation

Four bacterial strains were studied for their biofilm formation properties. *S. aureus* DSM 29134, *S. mutans* DSM 20523, and *E. faecalis* DSM 20,380 were bought from Leibniz Institute DSMZ-German Collection of Microorganisms and Cell Culture GmbH, and *E. coli* was isolated from a human fecal sample. All bacterial strains were cultured in trypticase soy yeast extract medium except *E. coli*, which was cultured in lysogeny broth (LB).

Biofilm formation of all strains was achieved using the same procedure. Bacterial cultures were grown overnight and centrifuged for 10 min at 8000× *g* to pellet the bacteria. Bacterial pellets were washed in 1× PBS, centrifuged for 10 min at 8000× *g*, and then diluted to OD600 = 0.25 with respective growth media. Five specimens of each implant type were used in bacterial adhesion experiments. Autoclaved specimens without saliva treatment and with saliva treatment were transferred in petri plates, and a bacterial suspension was added to immerse the specimens. Plates were then sealed with parafilm and incubated for 24 h at 37 °C.

### 2.4. Enumeration of Adhered Bacteria

Bacterial adhesion was enumerated by carefully removing the bacterial suspension and washing the specimens three times with 1× PBS. After washing of non-adhered bacteria, specimens were placed in six-well plates (one specimen/well), and 1 mL of 1× PBS was added into each well. Scraping of biofilm from specimens was performed using dental brush sticks (EAN 7630019902762). After detachment of biofilm, 1× PBS containing scraped bacteria was collected into Eppendorf tubes along with dental brush tips, followed by vigorous vortexing to remove scraped bacteria from the tips. Serial dilutions were generated for counting colony-forming units (CFU) in respective growth media, i.e., trypticase soy yeast extract medium agar and LB agar plates, so as to enumerate the bacteria in 1× PBS after detachment of bacteria from the biofilms. CFU count was calculated after incubation of the plates for 48 h at 37 °C.

### 2.5. Statistical Analysis

The *p*-values were calculated using a *t*-test (two-sample test assuming unequal variances). A *p*-value < 0.05 was considered statistically significant.

### 2.6. Preparation of Samples for Scanning Electron Microscopy (SEM)

A bacterial biofilm was allowed to form on the specimens under the growth conditions specified above for 24 h, followed by removal of nonadherent bacteria by washing with 1× PBS. The specimens with attached biofilms were prepared for SEM imaging by fixing with 1% glutaraldehyde and 4% paraformaldehyde in 0.1 M phosphate buffer, air-drying, and then sputter-coating with a 5 nm platinum layer. Imaging was conducted with a Sigma HD VP FE-SEM using an in-lens detector.

## 3. Results

### Bacterial Adhesion and Biofilm Formation

Figure 2 shows the photographed images of PEEK and titanium discs before and after bacterial biofilm formation, while Figure 3, Figure 4, Figure 5 and Figure 6 show the SEM images of *S. mutans*, *S. aureus*, *E. faecalis*, and *E. coli* on all implant material types studied. SEM was used to examine topography of the sample with high magnification. It was the preferred method for visualizing the biofilm since it not only provided information about the spatial structure of the biofilm but also detected the presence of extracellular matrix substances. However, conventional SEM has some limitations as it lacks vertical resolution and is deficient in providing data about the number of viable cells in the biofilm.

Figure 3 shows the typical structural morphology of *S. mutans*. The cells were cocci and arranged in short chains. In some areas of the biofilm, cells were present in aggregated form, and some areas showed the separation of cells from the biofilm.

Figure 4 shows the *S. aureus* morphology in the form of clusters, as well as single cells. Figure 4C presents the heavy growth of biofilm on PEEK materials as compared to Ti grade 2 and grade 5.

Figure 5 represents the characteristic growth of *E. faecalis* on all implant materials. They typically formed pairs or were arranged in short chains, as clearly visible in the images.

Figure 6 shows the traditional colony morphology of *E. coli* arranged individually and in pairs.

The bacterial adhesion test results are presented in Figure 7.

The PEEK plastic implant material showed greater bacterial adhesion as compared to titanium implant materials Ti grade 2 and Ti grade 5 by *S. aureus* (*p* = 0.0008) and *E. coli* (*p* = 0.05) in saliva-treated specimens, whereas, in the non-saliva-treated group, *S. mutans* (*p* = 0.03) and *E. faecalis* (*p* = 0.02) showed greater adhesion to PEEK than titanium specimens. *E. faecalis* also showed higher adhesion (*p* = 0.0003) to Ti grade 5 than PEEK in the saliva-treated group. No significant difference was found in *E. coli* between materials in the non-saliva-treated group.

In general, *S. mutans*, *S. aureus*, and *E. coli* showed greater attachment to PEEK as compared to titanium materials. However, on the other hand, *E. faecalis* showed greater adhesion to titanium specimens than PEEK.

Overall, the salivary coating on implant materials revealed significant elevation of bacterial attachment by *S. mutans*, *S. aureus*, and *E. faecalis*, whereas *E. coli* adherence showed no difference in saliva-treated and non-saliva-treated groups.

## 4. Discussion

In this study, we investigated four bacterial strains (*S. mutans*, *S. aureus*, *E. faecalis*, and *E. coli*) for their biofilm formation properties on two commonly used CAD–CAM materials used in patient-specific surgical reconstruction, PEEK and titanium. For titanium, both the grade 2 milled version and the grade 5 laser sintering version were selected to cover CAM techniques commonly used in maxillofacial surgery. Bacterial adhesion on implants was carried out in two groups: saliva-treated group and non-saliva-treated group. The salivary coating on implant materials revealed a significant elevation of bacterial attachment by *S. mutans*, *S. aureus*, and *E. faecalis*, whereas *E. coli* adherence showed no difference in the saliva-treated and non-saliva-treated groups. The PEEK plastic implant material showed greater bacterial adhesion compared to titanium implants (Ti grade 2 and Ti grade 5) except for *E. faecalis*, which showed greater adhesion to titanium implants. Similar results were found in a study conducted to evaluate the bacterial adhesion characteristics of PEEK and titanium cages for spinal infection which showed higher adhesion of *Staphylococcus* spp. to PEEK. The aim of this study was to compare and then recommend the use of a suitable implant material in osteomyelitis or spondylodiscitis [15].

PEEK is a broadly accepted and widely used material in spinal implants, trauma, orthopedic surgery, and cranioplasty [8,11]. The. biocompatibility, chemical stability, and radiolucency of PEEK make it a favorable implant material in bone reconstructions. Although PEEK has an elastic modulus similar to natural bone, it exhibits deficient osseointegration after implantation [16]. When compared in terms of mechanical differences between PEEK and titanium samples, the damage patterns of the models of titanium implants showed no deformation, but there was extensive damage to the fixation screws and the polyamide skull models [17]. While testing PEEK implants, PEEK itself fragmented into multiple pieces with only slight damage to the fixation screws and the polyamide skull models [17].

Implant site infections, adverse tissue reactions, capsule formation, and post-surgical wound infections are the main challenges in the implantation of biomaterials, causing very serious complications. The causative microbial agent depends on the type of surgery; however, the most common organisms isolated in surgical site infections are *S. aureus*, coagulase-negative *Staphylococci*, *Enterococcus* spp., and *E. coli* [18]. The significant bacterial species that cause infection of the spine such as osteomyelitis and spondylodiscitis are *S. aureus* (30–80%), *S. epidermidis* (10%), and Gram-negative bacteria (25%) including *E. coli* (5.6%) [15]. In the field of orthopedic arthroplasty, the incidence of hip prosthetic joint infection (PJI), a serious complication after hip arthroplasty, has been reported to be approximately 0.6–2.2% according to the Nordic Arthroplasty Register Association (NARA) dataset [19].

Webster et al. [20] compared silicon nitride (Si_3_N_4_), PEEK, and titanium implants in the reconstructions of rat calvarial defects with induced *S. epidermidis* infection. They found out that new bone formation was higher around titanium (26%) compared to PEEK (21%), while that with silicon nitride was even higher [20]. Another in vitro study designed to compare biofilm formation on silicon nitride, PEEK, and a titanium alloy (Ti6Al4V-ELI) using Gram-positive *S. epidermidis* and Gram-negative *E. coli* reported that both microbial species showed similar biofilm-forming trends. The highest density was found on PEEK, followed by titanium and silicon nitride [21].

Järvinen et al. (2019) published a retrospective study which included a cohort of 24 patients who underwent maxillofacial surgery using a PEEK patient-specific implant (PSI) with an infection rate of 8.3%. They reported wound dehiscence in two cases, but only one of the dehiscence wounds was infected [22]. Similar infection rates (14.3%) were also published by Alonso-Rodriguez et al. [11]. They reported a series of 14 patients with craniofacial defects who underwent reconstructions using a PEEK PSI, and two patients out of 14 showed a postoperative infection.

In Rochford et al.’s study, titanium and PEEK (commercially available) and their modified equivalents (surface-polished Ti and oxygen plasma-treated PEEK) were assessed in vitro and in vivo to evaluate their effect on the infection burden and immunological responses. They discovered that, once the implanted material was infected, the choice between titanium and PEEK showed no significant difference in the progression of the infection [23]. Suojanen et al. reported no difference in infection between patient-specific implants (manufactured either by milling from titanium monoblocks or by laser sintering from titanium powder) and conventional mini-plates (titanium) in the mandibular bilateral sagittal split osteotomy (BSSO) procedure. In the PSI group, wound infections were found in five patients out of 28, and the PSI was infected in two of them. In the mini-plate group, wound infections were developed in eight patients, along with plate infections in five out of 48 patients. Infection rates were not calculated in the PSI group [24]. Similar results were also found in a study carried out for the comparison between the PSI and conventional mini-plate group in a Le Fort I osteotomy. The infection rate was not statistically significant, and wound complications were rare in both groups. The groups did not differ statistically as a function of postoperative wound problems, infections (*p* = 0.500), or plate/screw removal (*p* = 0.668) [25].

Zhang et al. (2019) searched through 22 articles of PEEK cranioplasty reconstructions including 620 patient cases in their entirety, and they reported that the probability of an infection was 6.3%, with implant infection being the most frequent complication. Reoperation was needed with 7.3% of patients, and PEEK material needed to be removed from 4.8% of the patients [26]. Another study conducted on the use of PEEK in cranioplasty reconstructions in 65 patients presented an infection rate of 7.7% [27]. Punchak et al. also reviewed the use of PEEK in cranioplasty and created a meta-analysis comparing PEEK, autologous bone graft, and titanium implant. There was a trend toward lower complication rates when using PEEK, but with no statistical difference. Moreover, in this study, the main complication was infection; 11% of the 183 patients who underwent reconstructions using a PEEK implant developed postoperative infection. Infection rates with titanium mesh implants were between 0% to 11%, but comparing these rates was not statistically feasible [28]. In another review article, infection rates of different materials were assessed in cranioplasty and craniofacial reconstruction. Material selection in this article was broad, including also PEEK and titanium, and the average infection rates were 7.71% for titanium mesh, 8.31% for titanium plate, and 7.89% for PEEK. There were no statistically significant differences in infection rates between any of the material types studied [29]. In a retrospective study of 20 cranioplasty patients who received PEEK PSIs (15 patients) or titanium PSIs (five patients), one skin wound infection occurred, and wound dehiscence later emerged. In this case, the incision line was made across the area of the defect; therefore, the wound closure was made on the PEEK PSI instead of the bone as would be preferred, highlighting the importance of optimal wound location and closure [30].

More than 700 bacterial species are present in the oral cavity, and implant biomaterials are exposed to these wide varieties of bacteria and the varying pH of saliva. This results in biofilm formation on all exposed surfaces of biomaterials [31]. The contamination of biomaterial with saliva perioperatively and in the early stages of wound healing may lead to surgical infection. The presence of saliva and oral biofilm on the surface of the implant might favor the growth of pathogenic bacteria and negatively affect the important biomaterial properties needed for healing [32]. Regardless of the preventions and improvements in the clinical settings, surgical site infection (SSI) is still a serious unsolved clinical problem.

Some limitations existed in the present in vitro study. It is known that the in vitro conditions do not correspond to the clinical situations where several bacterial species affect bacterial biofilm formations at the same time, whereas, in the present study, only single bacterial strains were tested at once. The immune response and proteins involved in the clinical environment are not present in in vitro conditions. There are multiple proteins in saliva that contribute to the adherence of bacterial strains in vivo [33]; however, the filtration of saliva affects the total protein concentration and may result in the loss of certain proteins from the saliva.

## 5. Conclusions

On the basis of the present study, it can be concluded that the salivary coating had a more significant effect on PSI bacterial adhesion than the implant material type or CAM technique used. This suggests that an extraoral surgical approach must be considered, especially in immunocompromised patients when titanium or PEEK PSIs are used.

## Figures and Tables

**Figure 1 polymers-14-03862-f001:**
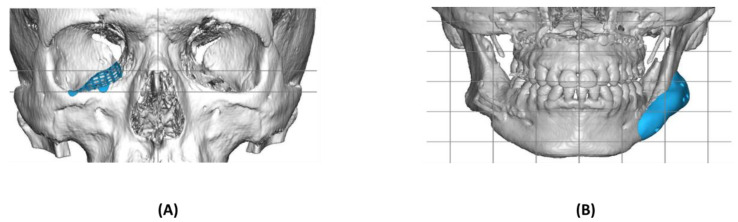
Virtual surgical planning and 3D simulation of patient-specific implant (Planmeca ProModelTM) for reconstruction of orbital floor fracture (**A**). The material used for the implant is titanium grade 23, with a thickness of 0.4 mm. Virtual surgical planning is used to maintain facial symmetry after bimaxillary surgery (**B**). Symmetry correction of the left mandible angle. Material PEEK, thickness of the plate: 0–7 mm.

**Figure 2 polymers-14-03862-f002:**
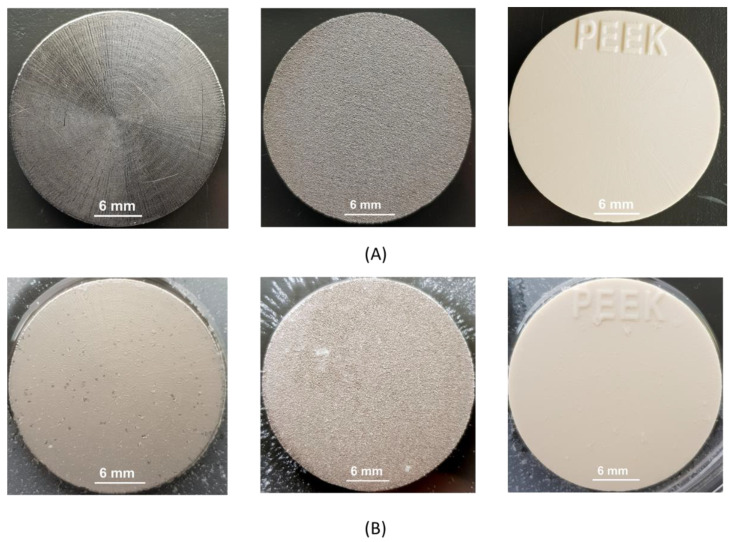
(**A**) Ti grade 2 (left), Ti grade 5 (center), and PEEK (right) without bacterial biofilm; (**B**) Ti grade 2 (left), Ti grade 5 (center), and PEEK (right) with *S. mutans* biofilm. Images were taken after 24 h of incubation in a bacterial suspension followed by washing of loosely attached bacterial cells.

**Figure 3 polymers-14-03862-f003:**
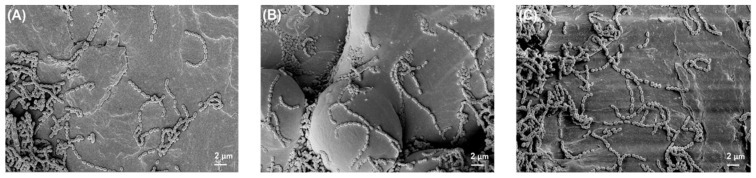
*S. mutans* biofilm on (**A**) Ti grade 2, (**B**) Ti grade 5, and (**C**) PEEK. All images were recorded at a high voltage of 5 kV with magnification = 3000×.

**Figure 4 polymers-14-03862-f004:**
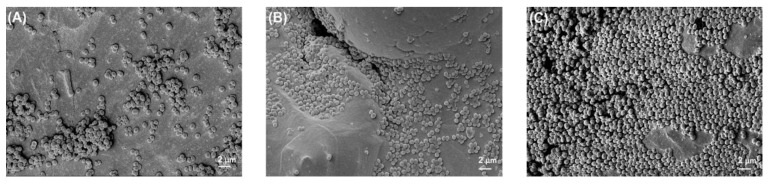
*S. aureus* biofilm on (**A**) Ti grade 2, (**B**) Ti grade 5, and (**C**) PEEK. All images were recorded at a high voltage of 5 kV and magnification = 3000×.

**Figure 5 polymers-14-03862-f005:**
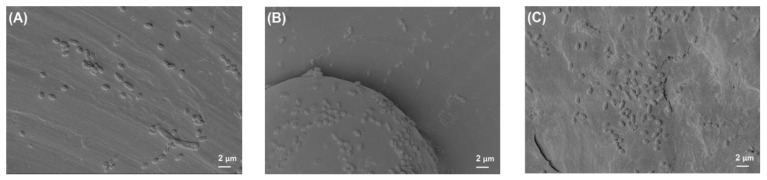
*E. faecalis* biofilm on (**A**) Ti grade 2, (**B**) Ti grade 5, and (**C**) PEEK. All images were recorded at high voltage of 5 kV and magnification = 3000×.

**Figure 6 polymers-14-03862-f006:**
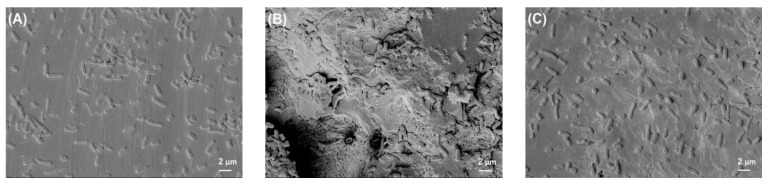
*E. coli* biofilm on (**A**) Ti grade 2, (**B**) Ti grade 5, and (**C**) PEEK. All images were recorded at high voltage of 5 kV and magnification = 3000×.

**Figure 7 polymers-14-03862-f007:**
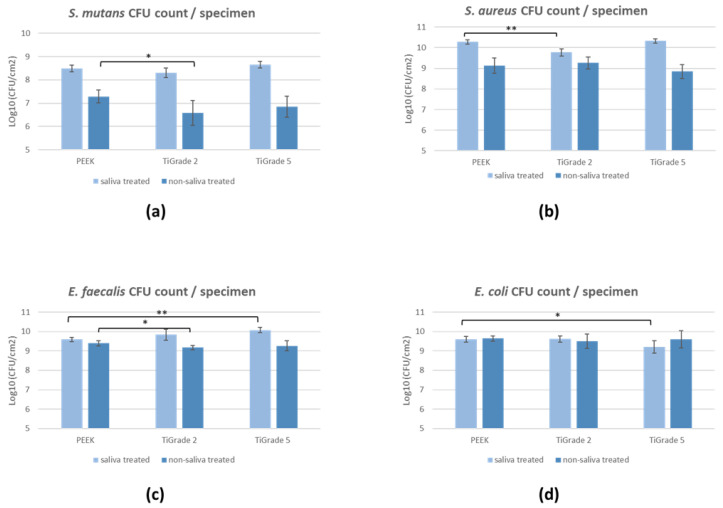
Colony-forming unit (CFU) count of (**a**) *S. mutans*, (**b**) *S. aureus*, (**c**) *E. faecalis*, and (**d**) *E. coli* on PEEK, Ti grade 2, and Ti grade 5. Statistical differences between materials are marked with lines; * *p* < 0.05, ** *p* < 0.001.

**Table 1 polymers-14-03862-t001:** Properties of PEEK and titanium.

	PEEK	Titanium
**Price for Single Piece Product**	~2000–2500 €	~2000–2500 €
Material cost (% of total costs)	30%	20%
Virtual surgical planning core service including engineering and tooling service (% of total costs)	70%	80%
Relative material costs	4	1
**Mechanical Characteristics**		
Compressive strength	+ ^1^	+
Bending stress	− ^2^	+
Puzzle-type design (multiple pieces)	+	−
Undermining structures possible	−	+
Radio-opacity	minor	significant
**Clinical Applications**	Puzzle-type structures (e.g., orbita)	Large defects
	Contour facial implants	Contour implants
	Cranial defects	Trauma
		Osteosynthesis

^1^ Positive aspects. ^2^ Negative aspects.

## Data Availability

Not applicable.

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
