# Peer review of "Comparison of Titanium and PEEK Medical Plastic Implant Materials for Their Bacterial Biofilm Formation Properties"

_polymers, 2022, doi:10.3390/polym14183862_

Round 1

Reviewer 1 Report

The authors investigated the biofilm formation of Staphylococcus aureus, Streptococcus mutans, Enterococcus faecalis, and Escherichia coli on titanium and PEEK medical implant materials, Two titanium specimens and 1 PEEK were studied, and the authors concluded that such salivary coating has a more significant effect on PSI bacterial adhesion than the implant material type or CAM-technique used, suggesting an extraoral surgical approach should be considered especially in the immuno-compromised patients when titanium or PEEK PSI are used. However, the authors are required to have more characterization techniques to support their conclusion rather than the only SEM images. 

Reviewer 2 Report

Manuscript numbered “polymers-1764463” has been reviewed:

The introduction needs some improvements.

Please add a suitable scale bar for figures.

Please add a comprehensive comparison for Ti and PEEK that includes price, mechanical properties, applications, etc.

Please add an economical study for using Ti and PEEK for fabricating the simple product.

Results have been just reported, please compare your finding with other research.

Fallowing papers are suggested for the introduction and result section:

Additive manufacturing a powerful tool for the aerospace industry

Optimization of FDM 3D printing parameters for high strength PEEK using the Taguchi method and experimental validation

Characterization of power demand and energy consumption for fused filament fabrication using CFR-PEEK

Three-dimensional printing technologies for dental prosthesis: a review

Open-source syringe extrusion head for shear-thinning materials 3D printing

Physical, thermal and tensile behaviour of 3D printed kenaf/PLA to suggest its usability for ankle–foot orthosis–a preliminary study
